# Prevalence of detectable viral load and its associated factors among adult patients receiving ART in Choma District, Zambia

**Macwani Mutukwa**[1,2]*, **Patrick Kaonga**[1], **Christine Mfula**[1,2], **Musa M. Mwansa**[2], **Benson M. Hamooya**[3]

**1** University of Zambia School of Public Health, Lusaka, Zambia, **2** Choma District Health Office, Choma, Zambia, **3** Mulungushi University School of Medicine and Health Sciences, Livingstone, Zambia

* cwanim@gmail.com

## Abstract

### Background

Africa accounts for two-thirds of the global HIV infection and a disproportionate burden is in sub-Saharan Africa. In 2017, the Zambian government launched the U=U campaign which has proven to be key in the prevention of HIV. However, there is a paucity of empirical evidence on the magnitude of detectable viral load in Choma district. This study aimed to estimate the proportion of detectable viral load and identify the associated factors among adults living with HIV receiving antiretroviral therapy (ART) in Choma District, Zambia.

### Methods

This was a cross-sectional study among adults aged 15 years and older on ART ≥12 months. Sociodemographic, clinical and laboratory data were collected through a structured questionnaire and data collection form for secondary data from medical records. Detectable Viral load (primary outcome) and Virological failure (secondary outcome) were defined as viral load (VL) >200cp/ml and VL >1000cp/ml respectively. The data collected was then analysed using STATA version XII. Descriptive statistics, chi-square test, Wilcoxon rank sum test, and logistic regression were the statistical methods used.

### Results

There was a total of 448 participants. The median (interquartile range (IQR)) age was 41 years (32, 49) of whom 284 (63.2%) were females. The prevalence of detectable and virological failure were 10.3% (n=46; 95% confidence interval (CI) 7.6, 13.5) and 5.4% (n=24; 95%CI 3.5, 7.9) respectively. In multivariable analysis, detectable VL was significantly associated with young age (16 – 24 years) (odds ratio (OR) 3.38; 95%CI 1.04, 10.94; p=0.042), no formal education (OR 3.32; 95%CI 1.06, 10.40; p=0.040), missing medication (OR 3.99; 95%CI 1.83, 8.73; p=0.001) and problem taking medication (OR 2.74; 95%CI 1.10; 6.84; p<0.030); while factors associated with virological failure were being in

**Data availability statement:** All relevant data are within the paper and its Supporting Information files.

**Funding:** The author(s) received no specific funding for this work.

**Competing interests:** The authors have declared that no competing interests exist.

age group 16 – 24 years (OR 7.28; 95%CI 1.62, 32.68, p = 0.009), male gender (OR 3.12; 95%CI 1.25, 7.76; p = 0.014), Missing taking medication (OR 8.28; 95%CI 2.59, 26.40; p < 0.001) and taking dolutegravir-based regimen with zidovudine/lamivudine backbone (OR 17.80 95% CI 2.29 - 132.31; p = 0.005).

## Conclusion

Detectable VL and virological failure were prevalent among adults receiving ART for ≥ 12 months and were significantly associated with sociodemographic and clinical factors. There is a need for targeted interventions, especially among young people and males to accelerate the attaining of the last 95 of the UNAIDS target; which is imperative in the prevention of HIV transmission. Qualitative research which aims to get an in-depth understanding of why men and young people do not attain optimal viral suppression is encouraged.

## Introduction

Human Immunodeficiency Virus (HIV) has remained one of the major public health concerns and it is associated with morbidity and mortality infecting over 75 million people and claiming over 32 million lives since its first record in 1983 [1]. According to the World Health Organization (WHO), approximately 39 million people were living with HIV globally as of 2022, with around 630,000 deaths annually attributed to HIV-related illnesses [2]. The impact of HIV/AIDS has been particularly profound in Africa, where nearly two-thirds of the global HIV-infected population resides, with almost 1 in every 25 adults living with HIV. This burden not only strains healthcare systems but also disrupts socio-economic structures [2,3]. However, the African region has made good strides towards the reduction of HIV new infection, morbidity and mortality.

In Zambia, HIV/AIDS is still the leading cause of morbidity and mortality [4]. Among adults aged 15 to 59 years, the HIV prevalence is 12.3% with an annual incidence of 0.66% [5]. The country targets to end HIV/AIDS by 2030 thus adopted implementation of the Sustainable Development Goal (SDG) number 6 in 2015, WHO guidelines on HIV management in 2013 and then UNAIDS target of 95-95-95; this implies that 95% of all people living with HIV should know their HIV status, 95% of all people with diagnosed HIV infection should receive sustained antiretroviral therapy (ART) and 95% of all people receiving antiretroviral therapy should have viral suppression [6]. However, most countries in resource-limited settings face challenges in achieving the last 95. In Zambia, for instance, the prevalence of viral load (VL) suppression among adult people with HIV is reported at 69% [7], 89.2% [5] and 86.4% [8]. To reduce the incidence of HIV and achieve HIV/AIDS epidemic control, in May 2019, the Zambian government launched the "Undetectable equals Untransmittable" campaign. Despite this historical launch, Zambia is still struggling with attaining the last 95 of the UNAIDS goals. This might be due to the lack of up-to-date data on the burden and factors related to detectable viral load and virological failure. Previous studies have identified several factors associated with suboptimal viral suppression, including young age [9] adherence to treatment [10–12], duration on ART and low CD4 count [13–16], TB co-infection and poor adherence [17], socioeconomic status [18,19], ethnicity and race [20–22], and alcohol consumption [23].

In sub-Saharan Africa (SSA), there is limited information on the country-specific data imperative for designing targeted evidence-based interventions aimed at achieving undetectable viral load among adults receiving ART, which is ultimately cardinal in reducing

HIV-related morbidities and mortalities [24]. To design targeted interventions, there is a need to generate current data on the magnitude and specific model/s of factors that influence the detectability of viral load among adult people with HIV receiving antiretroviral therapy in SSA. Therefore, this study aimed to determine the proportion of detectable VL and virological failure and identify associated factors among adults with HIV receiving ART in Choma district, Zambia.

## Materials and methods

### Study design and setting

We conducted a cross sectional study among HIV positive individuals accessing ART services in Choma district. Choma district is the provincial capital of Southern Zambia. It covers a total area of 5,210 square meters with a total population of about 180,673 during the 2010 census. Agriculture is the major economic activity for the locals with a few working for the government. There are two hospitals, Choma General Hospital and Macha Mission Hospital which offer specialised health services and 33 health facilities offering primary health care.

### Study population

The study was conducted from 25th January, 2023 to 24th February, 2023. We recruited 448 individuals attending routine ART clinics. Eligible individuals were adults aged ≥15 who had been on antiretroviral therapy for over one year and had at least two documented viral loads.

### Sample size estimation

Utilising the prevalence formula $n = (z)†p(1-p)/(d)†$, the minimum required sample size was determined to be 385 at a 95% confidence level and a 5% margin of error. After adjusting for a 20% non-response rate, the final sample size was calculated as 448

### Selection of participants and sampling methods

Purposive sampling was utilised to select the health facilities for the study, based on the number of HIV- positive clients receiving ART services at each health facility. The participants were selected from the seven (7) ART high volume facilities namely, Choma General Hospital, Shampande, Railway Surgery, Njase, Mbabala, Mapanza and Batoka clinics. A high-volume facility in Choma district was defined as a clinic that has over 500 people living with HIV actively taking their ART on their registers according to Zambia HIV/AIDS Prevention, Care and Treatment (ZPCT) of 2008. The number of participants in each health facility was calculated using the proportional-to-size ratio sampling method. At the facility level, systematic random sampling was used to select eligible participants and daily appointment registers as a sampling frame.

### Data collection plan and tools

A paper-based structured questionnaire and data collection form were used to collect sociode-mographic (age, sex, education level, marital status, employment or working status, monthly income, and religion), lifestyle (smoking and alcohol consumption status), laboratory (CD4 count and viral load) and clinical (perinatal HIV status, health facility, ART regimen, duration on ART, ever missed taking ART/medication, problem taking medication, side effects, and counseling levels) data from participants and medical health records (SmartCare, laboratory information system and patients' registers (facility CD4 and viral load registers)). To ensure validity and reliability, the research assistants had a one-day training on different

characteristics of the questionnaire and the objective of the project; the tool was also piloted on 20 eligible participants to assess the readability and clarity of the questions.

## Study outcomes

The Primary outcome, detectable HIV Viral load was defined as having HIV viral load of >200 copies/ml [25]. The secondary outcome, non-suppressed HIV viral load (virological failure) was defined as having HIV viral load > 1000copies/ml [26].

## Data management and storage

Questionnaires were kept in lockable cabinet where only the principal investigator and research assistants were able to access. De-identified data were entered onto Microsoft Excel for storage and cleaning purposes. Data were stored encrypted and only the principal investigator had access to it.

## Data analysis plan

STATA version XII (Stata Corp., College, Station, Texas, USA) was used for analyzing the data set. To understand the distribution of the data, descriptive statistics such as mean, median and interquartile range were used. Shapiro Wilk test and QQ plot were used to ascertain the normality of continuous variables. To determine the relationship between two categorical variables, the chi-square test was used. Wilcoxon rank-sum test was used to determine the statistical difference between the two medians. Bivariable logistic regression was used to select the variables to input in multivariable analysis. Multivariable logistic regression was used to ascertain factors associated with detectable and virological failure. A confidence interval (CI) of 95% and a 5% level of significance was used to determine significant variables.

## Ethical considerations

The research proposal, informed consent forms, assent forms and questionnaires in English were submitted for approval to the University of Zambia Biomedical Research Ethics Committee (UNZABREC: reference number **1666-2021**) and the Zambia National Health Research Authority (ZNHRA). Written informed consent or assent to participate in the survey was obtained from all participants. Participants were told that they were free to stop or withdraw participation at any time without any consequences.

## Results

### Descriptive characteristics

The study comprised 448 participants with a median (interquartile range (IQR) age of 41 years (32,49) of whom 284 (63.2%) were females. The majority of them had primary school education [179 (40.5%)], were not in employment [271 (60.9%)], and earned less than K1000 ($37) per month [285 (69.8%)]. A huge proportion of individuals were currently not smokers [420 (93.5%)] and not consuming alcohol [342 (76.5%)]. The majority of the clients had HIV type 1 [430 (95.9%)] with 16 (3.6%) having both HIV type 1 and 2. Among the participants, 7 (1.6%) contracted the virus perinatally. Almost all the participants were on a dolutegravir (DTG)-based regimen [448 (99.8]] with only one (1) client being on a protease inhibitor (PI). In terms of nucleoside reverse transcriptase inhibitors (NRTIs), the majority of the clients were on tenofovir disoproxil fumarate/lamivudine (TDF/3TC) (98%) and 1.6% zidovudine (AZT)/3TC. Most participants received TDF/3TC/DTG [439 (97.9%)]. The median (IQR) duration on ART was 84 months (48,132). Of the 109 participants with CD4 on file, the

median (IQR) CD4 count was 544 cells/ml (386, 731). A higher proportion of the participants reported not to have missed taking their medication [257 (57.5%)], not having problems taking medication on time [410 (92.1%)], and being provided with ordinary adherence counseling services during their clinical visits [430 (96.2%)]

## Relationships between outcomes (detectable VL and virological failure) and explanatory (sociodemographic, lifestyle and clinical) variables

**Detectable viral load.**  The prevalence of detectable viral load (VL) was 10.3% (n = 46; 95% confidence interval (CI) 7.6 - 13.5). Males had a significantly higher prevalence of detectable VL (52.2% vs. 35.1%, p = 0.023). Those without formal education had a significantly higher proportion of detectable VL (37.8% vs. 14.8%, p = 0.001). Individuals born from HIV-positive parents had a significantly higher prevalence of detectable VL (6.5% vs. 1.0%, p = 0.004.) Participants from Batoka Rural Health center and Choma Railway Surgery had higher detectable VL prevalence. Individuals who missed taking medication had a much higher proportion of detectable VL (71.7% vs. 39.2%, p < 0.001). Those who faced difficulties in taking medication had a higher proportion of detectable VL (28.3% Vs 5.5% p < 0.001). Participants on enhanced adherence counselling had a higher proportion of detectable VL (32.6% vs. 0.5%, p < 0.001), see Table 1.

**Virological failure.**  The study found a 5.4% (n = 24; 95%CI 3.5, 7.9) prevalence of virological failure. A significantly higher proportion of males (62.5% vs. 35.2%, p = 0.007) and those without formal education (41.6% Vs 16.7%, p = 0.005) had virological failure. Among participants who perinatally acquired HIV, a higher proportion of them had virological failure, 8.3% vs. 1.2%; p = 0.006. The proportion of virological failure was significantly higher at Batoka Rural Health Center (33.3% vs. 7.8%, p < 0.001) and Choma Railway Surgery (33.3% vs. 17.0%, p < 0.001). Participants on AZT/3TC/DTG regimen were more likely to have virological failure, 8.3% vs. 1.0%; p = 0.050. Individuals with a shorter duration on ART had virological failure, 60 months vs. 84 months; p = 0.044. A higher prevalence of participants who reported to have ever missed taking medication had virological failure, 83.3% vs. 40.3%, p < 0.001. Equally, the majority of individuals who reported having a problem taking medication had virological failure, 45.8% vs. 5.7%; p < 0.001. A significantly higher proportion of participants who received enhanced adherence counseling experienced virological failure (62.5% compared to 0.5%, p < 0.001), see Table 1.

## Logistic regression analysis of sociodemographic and lifestyle factors associated with detectable VL and virological failure

**Detectable viral load.**  In unadjusted analysis, the younger individuals (16-24 years) had a significantly 2.98 higher odds of having detectable compared with those aged 45-76 years. Males compared to females, were significantly 2.02 times more likely to have detectable VL. Individuals with no formal schooling had a significantly 2.99 increased odds of having detectable compared to those with college/university education. In adjusted analysis, participants aged 16 to 24 years were 3.38 times more likely to have detectable VL compared with those aged 45 to 76 years, p = 0.042. Participants with no formal education were significantly 3.32 times more likely to have a detectable viral load compared to those with a college or university education, see Table 2.

**Virological failure.**  In unadjusted analysis, individuals aged 16 to 24 years had a 4.85 increased odds of having virological failure compared with those aged 45 to 76 years, p = 0.024. Males had a significantly 3.04 increased odds of having virological failure compared with their female counterparts. In adjusted analysis, participants aged 16 to 24 years were

**Table 1. Prevalence and association of detectable VL and virological failure with other variables of participants on ART.**

| Characteristics | Detectable Viral Load | | | | Virological Failure | | | |
|---|---|---|---|---|---|---|---|---|
| | N = 449 | Yes, 46 (10.3%) | No, 402 (89.7%) | p-value | N = 448 | Yes, 24 (5.4%) | No, 424 (94.6%) | p-value |
| **Age** (years)** | 449 | | | | 448 | | | |
| 45-76 | | 15 (32.6) | 160 (39.8) | 0.154 | | 5 (20.8) | 170 (40.1) | 0.113 |
| 35-44 | | 14 (30.4) | 128 (31.9) | | | 8 (33.3) | 134 (31.6) | |
| 25 - 34 | | 10 (21.7) | 89 (22.1) | | | 7 (29.2) | 92 (21.7) | |
| 16 - 24 | | 7 (15.2) | 25 (6.2) | | | 4 (16.7) | 28 (6.6) | |
| **Sex**** | 448 | | | **0.023** | 448 | | | **0.007** |
| Female | | 22 (47.8) | 261 (64.9) | | | 9 (32.5) | 264 (64.6) | |
| Male | | 24 (52.2) | 141 (35.1) | | | 15 (62.5) | 150 (35.4) | |
| **Education Level **** | 441 | | | **0.001** | 441 | | | **0.005** |
| College/University | | 5 (11.1) | 52 (13.1) | | | 4 (16.7) | 53 (12.7) | |
| Secondary school | | 12 (26.7) | 119 (30.0) | | | 6 (25.0) | 125 (30.0) | |
| Primary school | | 11 (24.4) | 167 (42.1) | | | 4 (16.7) | 174 (41.6) | |
| No formal schooling | | 17 (37.8) | 59 (14.8) | | | 10 (41.6) | 66 (15.7) | |
| **Marital Status **** | 446 | | | 0.787 | 446 | | | 0.840 |
| Yes | | 27 (58.7) | 243 (60.8) | | | 15 (62.5) | 255 (60.4) | |
| No | | 19 (41.3) | 157 (39.2) | | | 9 (37.5) | 167 (39.6) | |
| **Working in the last 12 months **** | 444 | | | 0.756 | 444 | | | 0.546 |
| Yes | | 19 (41.3) | 155 (38.9) | | | 8 (33.3) | 166 (39.5) | |
| No | | 27 (58.7) | 234 (61.1) | | | 16 (66.7) | 254 (60.5) | |
| **Monthly income > K1000 **** | 407 | | | 0.63 | 407 | | | 0.908 |
| Yes | | 15 (33.3) | 108 (29.8) | | | 7 (29.2) | 116 (30.3) | |
| No | | 30 (66.7) | 254(70.2) | | | 17 (70.8) | 267 (69.7) | |
| **Religion**** | 434 | | | 0.965 | 434 | | | 0.864 |
| Catholic | | 7 (15.2) | 60 (15.5) | | | 4 (16.7) | 63 (15.4) | |
| Protestants | | 39 (84.8) | 328 (84.5) | | | 20 (83.3) | 347 (84.6) | |
| **Living Alone**** | 442 | | | 0.328 | 442 | | | 0.238 |
| Yes | | 1 (2.2) | 22 (5.6) | | | 0 (0) | 23 (5.5) | |
| No | | 45 (97.8) | 374 (94.4) | | | 24 (100) | 395 (94.5) | |
| **Number of People living with > 18years*** | 442 | 3 (2,5) | 3 (2,4) | 0.225 | 442 | 3 (2,6) | 3 (2,4) | 0.349 |
| **Currently Smoke**** | 448 | | | 0.989 | 448 | | | 0.217 |
| Yes | | 3 (6.5) | 26 (6.5) | | | 3 (12.5) | 26 (16.1) | |
| No | | 43 (93.5) | 376 (93.5) | | | 21 (87.5) | 395 (93.9) | |
| **Currently consume alcohol**** | 447 | | | 0.943 | 447 | | | 0.500 |
| Yes | | 11 (23.9) | 94 (23.4) | | | 7 (29.2) | 98 (23.2) | |
| No | | 35 (76.1) | 307 (76.6) | | | 17 (70.8) | 325 (76.8) | |
| **Perinatal HIV**** | 448 | | | **0.004** | 448 | | | |
| Yes | | 3 (6.5) | 4 (1.0) | | | 2 (8.3) | 5 (1.2) | **0.006** |
| No | | 43 (93.5) | 398 (99.0) | | | 22 (91.7) | 419 (98.8) | |
| **Health Facility**** | 449 | | | **0.005** | 448 | | | **<0.001** |
| Choma General Hospital | | 12 (26.1) | 124 (30.8) | | | 3 (12.5) | 133 (31.2) | |
| Batoka Rural Health Center | | 9 (19.6) | 32 (8.0) | | | 8 (33.3) | 33 (7.7) | |
| Choma Railway Surgery | | 15 (32.6) | 65 (16.2) | | | 8 (33.3) | 74 (17.4) | |
| Shampande Urban Clinic | | 6 (13.0) | 114 (28.3) | | | 4 (16.7) | 116 (27.2) | |
| Mapanza Rural Health center | | 1 (2.2) | 28 (7.0) | | | 0 (0.0) | 29 (6.8) | |
| Njase Peri-urban clinic | | 3 (6.5) | 32 (8.0) | | | 1 (1.2) | 34 (8.0) | |
| Mbabala rural Health center | | 0 (0.0) | 7 (1.7) | | | 0 (0.0) | 7 (1.7) | |

*(Continued)*

**Table 1.**  (Continued)

| Characteristics | Detectable Viral Load | | | | Virological Failure | | | |
|---|---|---|---|---|---|---|---|---|
| | N = 449 | Yes, 46 (10.3%) | No, 402 (89.7%) | p-value | N = 448 | Yes, 24 (5.4%) | No, 424 (94.6%) | p-value |
| **CD4 count** (cells/ml)* | 109 | 445 (320, 680) | 549 (390, 732) | 0.429 | 109 | 445 (320, 680) | 549 (386, 733) | 0.523 |
| **ART Regimen**** | 447 | | | 0.43 | 447 | | | 0.050 |
| TDF/3TC/DTG | | 44 (95.7) | 394 (98.3) | | | 22 (91.7) | 416 (98.4) | |
| TAF/3TC/DTG | | 0 (0) | 1 (0.2) | | | 0 (0) | 1 (0.2) | |
| AZT/3TC/LPVr | | 0 (0) | 1 (0.2) | | | 0 (0) | 1 (0.2) | |
| AZT/3TC/DTG | | 2 (4.3) | 4 (1.1) | | | 2 (8.3) | 4 (1.0) | |
| ABC/3TC/DTG | | 0 (0) | 1 (0.2) | | | 0 (0) | 1 (0.2) | |
| **Duration on ART** (months)* | 447 | 62 (48 120) | 84 (48 132) | 0.103 | 447 | 60 (38,104) | 84 (48,132) | **0.044** |
| **Ever missed taking medication**** | 446 | | | **<0.001** | 446 | | | **<0.001** |
| Yes | | 33 (71.7) | 157 (39.2) | | | 20 (83.3) | 170 (40.3) | |
| No | | 13 (28.3) | 243 (60.8) | | | 4 (16.7) | 252 (59.7) | |
| **Problems with taking medication**** | 444 | | | **<0.001** | 444 | | | **<0.001** |
| Yes | | 13 (28.3) | 22 (5.53) | | | 11 (45.8) | 24 (5.7) | |
| No | | 33 (71.7) | 396 (94.5) | | | 13 (54.2) | 396 (94.3) | |
| **Side effects**** | 446 | | | 0.606 | 446 | | | 0.217 |
| Yes | | 1 (2.2) | 5 (1.3) | | | 1 (4.2) | 5 (1.2) | |
| No | | 45 (97.8) | 371 (98.5) | | | 23 (95.8) | 417 (98.8) | |
| **Counselling levels**** | 446 | | | **<0.001** | 446 | | | **<0.001** |
| Ordinary | | 31 (67.4) | 398 (99.5) | | | 9 (37.5) | 420 (99.5) | |
| Enhanced | | 15 (32.6) | 2 (0.5) | | | 15 (62.5) | 2 (0.5) | |

Note:

* data is presented as median (interquartile range),

** data presented as frequency (percent), CD4 cluster of differentiation 4, ART antiretroviral therapy, NRTI nucleoside reverse transcriptase inhibitors, AZT Zidovudine, ABC Abacavir, 3TC Lamivudine, TDF Tenofovir, TAF Tenofovir Alafenamide, PI protease inhibitor, LPV/R lopinavir/ ritonavir INSTI integrase strand transfer inhibitors DTG dolutegravir, bold p- values show variables statistically significant <0.05.

significantly 7.28 times more likely to have virological failure. Males compared to females, had a significantly 3.12 increased chance of having virological failure, see Table 2.

## Logistic regression analysis of clinical factors associated with detectable VL and virological failure

**Detectable viral load.**  In unadjusted analysis, participants from Batoka Rural Health Center and Choma Railway Surgery were significantly 2.90 and 2.38 times more likely to have detectable VL compared with those from Choma General Hospital, respectively. Individuals who perinatally acquired HIV had a significantly 6.94 increased odds of having detectable VL. Participants who reported having had missed taking medication were 3.93 times more likely to have detectable VL, p < 0.001. similarly, individuals who reported to have a problem taking the medication were significantly associated with 6.73 increased odds of having detectable VL. In adjusted analysis, individuals who reported having had missed taking medication were significantly associated with a 3.99 increased chance of having detectable VL. Participants who reported that they had a problem taking medication had 2.74 increased odds of having detectable VL, p = 0.030. See Table 3.

**Virological failure.**  In unadjusted analysis, participants from Batoka Rural Health Center and Choma Railway Surgery Clinic had a significantly 10.75 and 4.93 higher odds of having virological failure, respectively. Individuals who perinatally acquired HIV had 7.60 increased

**Table 2. Detectable VL and virological failure and sociodemographic characteristics.**

| Characteristics | Detectable Viral load | | | | | | Virological failure | | | | | |
|---|---|---|---|---|---|---|---|---|---|---|---|---|
| | Unadjusted analysis | | | Adjusted analysis | | | Unadjusted analysis | | | Adjusted analysis | | |
| | OR | 95% CI | p-value | OR | 95% CI | p-value | OR | 95% CI | p-value | OR | 95% CI | p-value |
| **Age**[**] | | | | | | | | | | | | |
| 45-76 | Ref | | | Ref | | | Ref | | | Ref | | |
| 35-44 | 1.17 | 0.54 - 2.51 | 0.693 | 1.24 | 0.54 - 2.88 | 0.612 | 2.02 | 0.65 - 6.35 | 0.224 | 2.83 | 0.85 - 9.47 | 0.091 |
| 25 - 34 | 1.19 | 0.52 - 2.78 | 0.673 | 1.11 | 0.43 - 2.89 | 0.815 | 2.59 | 0.79 - 8.38 | 0.113 | 3.34 | 0.94 - 11.81 | 0.062 |
| 16 - 24 | 2.98 | 1.11 - 8.05 | **0.031** | 3.38 | 1.04 - 10.94 | **0.042** | 4.85 | 1.23 - 19.19 | **0.024** | 7.28 | 1.62 - 32.68 | **0.009** |
| **Sex** | | | | | | | | | | | | |
| Female | Ref | | | Ref | | | | | | Ref | | |
| Male | 2.02 | 1.09 - 3.73 | **0.025** | 1.88 | 0.94 - 3.75 | 0.074 | 3.04 | 1.30 - 7.12 | **0.01** | 3.12 | 1.25 - 7.76 | **0.014** |
| **Education Level** | | | | | | | | | | | | |
| College/University | Ref | | | Ref | | | | | | | | |
| Secondary school | 1.05 | 0.35 - 3.13 | 0.932 | 0.70 | 0.22 - 2.28 | 0.56 | 0.60 | 0.17 - 2.34 | 0.497 | | | |
| Primary school | 0.69 | 0.23 - 2.06 | 0.501 | 0.59 | 0.19 - 1.86 | 0.37 | 0.3 | 0.07 - 1.26 | 0.101 | | | |
| No formal schooling | 2.99 | 1.03 - 8.69 | **0.043** | 3.32 | 1.06 - 10.40 | **0.04** | 2.00 | 0.59 - 6.76 | 0.261 | | | |
| **Marital Status** | | | | | | | | | | | | |
| No | | | | | | | | | | | | |
| Yes | 0.92 | 0.49 - 1.71 | 0.787 | | | | 1.09 | 0.47 - 2.55 | 0.84 | | | |
| **Currently Smoked** | | | | | | | | | | | | |
| Yes | | | | | | | | | | | | |
| No | 0.99 | 0.29 - 3.41 | 0.989 | | | | 0.46 | 0.13 - 1.63 | 0.228 | | | |
| **Alcohol use** | | | | | | | | | | | | |
| Yes | | | | | | | | | | | | |
| No | 0.97 | 0.48 - 1.99 | 0.943 | | | | 0.73 | 0.29 - 1.82 | 0.502 | | | |

OR odds ratio, CI confidence interval at 95%, CD4 cluster of differentiation four (4), ART anti retro-viral therapy, AZT Zidovudine, ABC Abacavir, 3TC Lamivudine, TDF Tenofovir, TAF Tenofovir Alafenamide, PI protease inhibitor, LPV/R lopinavir/ ritonavir INSTI integrase inhibitors DTG dolutegravir, bold p- values show variables statistically significant <0.05.N/A variable removed from analysis due to lack of outcome of interest, Ref - reference category.

odds of having virological failure. Participants on AZT/3TC/DTG regimen were 9.45 times more likely to have virological failure, p = 0.012. individuals who reported to have ever missed taking medication had 7.41 increased chance of having virological failure, p < 0.001. Similarly, those who reported to have a problem taking the medication had 13.36 increased odds of having virological failure, p < 0.001. In adjusted analysis, participants from Choma Railway Surgery Clinic were significantly 8.1 times more likely to have virological failure. Individuals on DTG with AZT/3TC backbone had a significantly 17.80 increased odds of having a virological failure. Participants who reported to have ever missed taking medication were 3.54 times more likely to have virological failure, p = 0.002. Equally, those who reported to have a problem taking the medication had 4.39 increased odds of having virological failure, p = 0.012, Table 3.

## Discussion

This study aimed to estimate the burden of and factors associated with detectable VL and virological failure among adults with HIV on antiretroviral therapy (ART) in Choma district of Zambia. Both conditions were common and positively associated with younger age, missing medication doses, and experiencing challenges in taking medication. Additionally, virological

**Table 3. Detectable VL and virological and clinical characteristics.**

| Characteristics | Detectable Viral load | | | | | | Virological failure | | | | | |
|---|---|---|---|---|---|---|---|---|---|---|---|---|
| | Unadjusted analysis | | | Adjusted analysis | | | Unadjusted analysis | | | Adjusted analysis | | |
| | OR | 95% CI | p-value | OR | 95% CI | p-value | OR | 95% CI | p-value | OR | 95% CI | p-value |
| **Health facility** | | | | | | | | | | | | |
| Choma General Hospital | Ref | | | | | | Ref | | | | | |
| Batoka Rural Health Center | 2.90 | 1.13 - 7.49 | **0.027** | 1.29 | 0.41 – 4.05 | 0.657 | 10.75 | 2.70 - 42.74 | **0.001** | 3.94 | 0.81 – 19.16 | 0.089 |
| Choma Railway Surgery | 2.38 | 1.05 - 5.39 | **0.037** | 1.74 | 0.40 – 7.57 | 0.458 | 4.93 | 1.27 - 19.15 | **0.021** | 8.15 | 1.60 – 41.46 | **0.011** |
| Shampande Urban Clinic | 0.54 | 0.19 - 1.49 | 0.238 | 1.12 | 0.35 – 3.50 | 0.845 | 1.53 | 0.34 - 6.97 | 0.584 | 1.92 | 0.27 – 13.51 | 0.513 |
| Mapanza Rural Health center | 0.36 | 0.05 - 2.96 | 0.348 | 0.46 | 0.04 – 5.52 | 0.545 | N/A | | | | | |
| Njase Peri-urban clinic | 0.96 | 0.26 - 3.64 | 0.962 | 2.67 | 0.55 – 12.95 | 0.222 | 1.3 | 0.13 - 12.93 | 0.821 | 2.93 | 0.20 – 42.01 | 0.43 |
| Mbabala rural Health center | n/a | | | | | | N/A | | | | | |
| **Perinatal HIV** | | | | | | | | | | | | |
| No | Ref | | | | | | Ref | | | | | |
| Yes | 6.94 | 1.50 - 32.05 | **0.013** | 2.02 | 0.26 – 15.72 | 0.499 | 7.6 | 1.39 - 41.49 | **0.019** | 3.27 | 0.24 - 36.58 | 0.385 |
| **CD4 count (cells/ml)** | 0.99 | 0.99 - 1.00 | 0.341 | | | | 0.99 | 0.99 - 1.00 | 0.528 | | | |
| **ART Regimen** | | | | | | | | | | | | |
| TDF/3TC/DTG | Ref | | | | | | Ref | | | Ref | | |
| TAF/3TC/DTG | n/a | | | | | | N/A | | | N/A | | |
| AZT/3TC/LPVr | 4.48 | 0.79 - 25.15 | 0.089 | | | | N/A | | | N/A | | |
| AZT/3TC/DTG | n/a | | | | | | 9.45 | 1.64 - 54.44 | **0.012** | 17.80 | 2.39 - 132.31 | **0.005** |
| ABC/3TC/DTG | n/a | | | | | | N/A | | | N/A | | |
| **Duration on ART (months)** | 0.99 | 0.99 - 1.00 | 0.152 | | | | 0.99 | 0.98 - 1.00 | 0.097 | | | |
| **Ever missed taking medication** | | | | | | | | | | | | |
| No | Ref | | | Ref | | | Ref | | | Ref | | |
| Yes | 3.93 | 2.01-7.69 | **<0.000** | 3.99 | 1.83 - 8.73 | **0.001** | 7.41 | 2.45 - 22.07 | **<0.001** | 3.54 | 1.59 - 7.85 | **0.002** |
| **Problems with taking medication** | | | | | | | | | | | | |
| No | Ref | | | Ref | | | Ref | | | Ref | | |
| Yes | 6.73 | 3.11 - 14.58 | **<0.000** | 2.74 | 1.10 - 6.84 | **0.03** | 13.36 | 5.67 - 34.42 | **<0.001** | 4.39 | 1.38 – 13.95 | **0.012** |
| **Side effects** | | | | | | | | | | | | |
| Yes | Ref | | | | | | Ref | | | | | |
| No | 0.57 | 0.07 - 4.98 | 0.611 | | | | 0.28 | 0.03 - 2.46 | 0.248 | | | |

OR odds ratio, CI confidence interval at 95%, CD4 cluster of differentiation four (4), ART anti retro-viral therapy, AZT Zidovudine, ABC Abacavir, 3TC Lamivudine, TDF Tenofovir, TAF Tenofovir Alafenamide, PI protease inhibitor, LPV/R lopinavir/ ritonavir INSTI integrase inhibitors DTG dolutegravir, bold p- values show variables statistically significant < 0.05.N/A variable removed from analysis due to lack of outcome of interest, Ref - reference category.

failure was associated with male gender, receiving ART at a clinic, and specific ART regimens, while detectable VL was linked to the absence of formal education.

The prevalence of virological failure was lower compared to previous findings from Eswatini, Lesotho, Malawi, Zambia, Zimbabwe [27], and Uganda [28], where prevalences exceeded 10%. Similarly, the proportion of detectable VL was lower in comparison to a study conducted in South Africa [29]. This could be attributed to the introduction of the 'Treat and Treat' strategy in 2017, as the other studies included data prior to its implementation. Early initiation of newly diagnosed HIV-positive individuals on ART has proven to immensely reduce the viral load to undetectable limits within 4 to 12 weeks [30]. The improvement in undetectable viral load among recipients of care is a great milestone achieved in the management of HIV/ AIDS regarding the use of treatment as prevention (U = U) [31] and achievement of epidemic control by 2030.

Younger individuals (aged 16-24years) were more likely to have detectable VL and VF unlike the older ones (aged 45-76 years). This aligns with previous studies in Africa, which shows that the transition from paediatrics to adult ART is associated with adherence challenges and higher VL [32–34]. Adolescents and young adults face unique challenges such as treatment fatigue, mental health issues, stigma, and inadequate social support, all of which hinder treatment adherence [29,35]. Hence, the need for targeted inventions such as peer-to-peer support and better transition strategies to address the high prevalence of detectable VL in the age group.

Males were more likely to have a detectable viral load and virological failure than females. The finding is consistent with previous research, which shows that men generally have poor health-seeking behavior [36,37]. Men seek care later in disease progression and may be less engaged with health care and remain transmitting to female partners through condomless sex [37,38–41]. Additionally, some studies suggest that hormones contribute to gender disparities in viral suppression, with female sex hormones, such as estradiol and progesterone, playing a role in enhancing viral suppression [42]. Improving HIV management and adherence in the male population would require the establishment of men's clinics and prioritizing them in ART clinics [43].

Participants with lower education levels were more likely to have detectable VL which aligns with previous studies in Uganda [28] and Ghana [44]. This may be attributed to the effects of low socio-economic status, limited health literacy and access to information [45,46]. Educated individuals tend to have better understanding of ART which may lead to improved adherence resulting in having undetectable viral load [47].

Adherence to ART was also significantly associated with virological failure and detectable VL, participants with problems taking their medication and reported missing drugs were more likely to have detectable viral load. This is consistent with previous studies in Guatamela which also underscored the importance of adherence [48]. Interruption in treatment over a short period can result in virological rebound, and rebounds tend to be accompanied by resistant mutations [49,50]. The study revealed that participants on enhanced adherence counselling were more likely to have a detectable viral as compared to those receiving standard counselling sessions. This is due to ART clients with high viral load being offered enhanced adherence counselling according to the Zambian consolidated guidelines on HIV management [51]. Enhance adherence counselling (EAC) is a targeted counselling strategy designed to help clients identify individual barriers to adherence and develop strategies to improve viral suppression within three months [52]. Additionally, health facility was found to be significantly associated with detectable viral, where participants from Choma Railway Surgery Clinic had increased odds of having virological failure/detectable viral load. This could be due to system factors such as human resource challenges, client management systems, and general administration of facilities [53].

Individuals taking a combination of AZT/3TC/DTG had an increased likelihood of having a virological failure. According to Zambia consolidated guidelines of 2022, clients failing on TAF/3TC/DTG and TDF/3TC/DTG should alternatively be switched to AZT/3TC/DTG on the second line [51]. Studies have shown that with time, some individuals lose susceptibility to TDF thereby resulting in mutations called thymidine analogue mutations (TAMS) but however, maintain susceptibility to AZT/3TC [54].

## Limitations and strengths of the study

This was a cross-sectional study, therefore we could not determine the temporal relation between detectable VL or virological failure and the explanatory variables. Some information

may have been inaccurate, especially the self-reported questions on missing taking medication without pill count verification. However, this study was able to quantify detectable VL, a concept that is very cardinal in U = U, and identified the associated factors. This area is understudied in our country and most low-and-middle-income countries.

## Conclusion

The findings of our study demonstrated that detectable viral load is prevalent among adult clients on ART. The factors associated with detectable viral load were young age (16-24 years), male gender, having no formal education, and missing taking drugs. There is a need to enhance the interventions meant to improve undetectable viral load among ART clients, especially the ones targeted at young people, males, and those having difficulties taking medication. Furthermore, the study demonstrated the attainment of the third 95 of the UNAIDS strategy toward elimination of new HIV infection by 2030. However, there is still a need to attain optimal viral suppression among males, young people, and those who miss taking medication.

## Supporting information

**S1 minimal dataset. The dataset on which all the study findings reported in the article are based.**
(XLSX)

## Acknowledgments

The authors appreciate PEPFAR, CDC and other implementing partners that support HIV programming in southern province, the district health director, the health facility personnel and the research assistants Ailess Mbewe, Kelvin Mweemba, Vigilance Simakani, Charlotte Katongo, Diana Hanagwete, Silvia Mbuyoti and Davies Lungenda

## Author contributions

**Conceptualization:** Macwani Mutukwa.

**Data curation:** Macwani Mutukwa.

**Formal analysis:** Macwani Mutukwa, Benson M. Hamooya.

**Investigation:** Macwani Mutukwa, Benson M. Hamooya.

**Methodology:** Macwani Mutukwa, Benson M. Hamooya.

**Project administration:** Macwani Mutukwa.

**Supervision:** Patrick Kaonga, Benson M. Hamooya.

**Writing – original draft:** Macwani Mutukwa.

**Writing – review & editing:** Macwani Mutukwa, Christine Mfula, Musa M. Mwansa, Benson M. Hamooya.

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
