## [Decision Letter · Decision Letter 0]

1 Oct 2024

PONE-D-24-18373Prevalence and factors associated with detectable viral load among adult patients receiving ART in Choma District, ZambiaPLOS ONE

Dear Dr. Mutukwa,

Thank you for submitting your manuscript to PLOS ONE. After careful consideration, we feel that it has merit but does not fully meet PLOS ONE’s publication criteria as it currently stands. Therefore, we invite you to submit a revised version of the manuscript that addresses the points raised during the review process.This is an important study that aims to describe factors associated with VL suppression among clients receiving ART in Choma District, Zambia. Viral suppression is critical requirement that will facilitate AIDS-free generation efforts by 2030, hence the study is timely. Unfortunately, the study is not presented in a concise manner that will aid comprehension by readers. Additionally, the methodology lacks details to aid reproducibility. Therefore, the authors are requested to please review carefully the following comments in addition to comments provided by the 3 reviewers below. Failure to respond to tall comments will delay or result in rejection of this paper in the next round of revision. Therefore, kindly restructure the paper accordingly and ensure conciseness and clarity.

Data availability – stated “some restrictions will apply” please explain the level of restrictions and how the data can be accessed.Title: Prevalence and factors associated with detectable viral load among adult patients receiving ART in Choma District, Zambia. “Prevalence” is confusing in the title, is it referring to prevalence of factors or prevalence of detectable VL? Please make it clear, for example “Prevalence of Detectable VL and Associated Factors among Adult Patients Receiving ART in Choma District, Zambia”.Introduction: Good epidemiologic context provided. However, no reference made to factors influencing VL suppression. Consider restructuring the section using 95:95:95 cascade and about 50% of the section should focus on factors influencing VL suppression, including gaps in Zambia and study region.Methods:Study design: is it primary data collection or secondary data analysis?Selection and sampling: 448 clients were selected. Please describe sampling criteria, method and sample size estimation. Also describe accurately “selection based on proportional to size”.Data collection and storage: was the data collected using paper-based questionnaire or electronically? What are the variables of interest and their definition?
Results: not structured and presented without keeping the study objectives in mind. Of course, the study objectives were not well defined in the methods, and this may be the reason for the long and unstructured results section. The section is very lengthy and difficult to follow. No need to mention all findings. Focus on major findings that are elated to the objectives. Try to make the text for this section like maximum of 1 page. The tables should be self-explanatory. The tables are also too long and confusing. Focus on key findings and avoid repetition.Table 1: should be “Sociodemographic characteristics and VL suppression of participants from selected health facilities in Choma, Zambia”. The table should end at “perinatal HIV” because the variables that follow are not sociodemographic and should be moved to another table.Table 2: Please avoid repetition. Harmonize table 1 and 2 based on the comment above. Make Table 2 to focus on “VL suppression by site and clinical characteristics”Tables 3 and 4: please harmonize these tables into 1 table and be concise by focusing on the primary intention of the table.  
Discussion: too long. Please be concise and focus on key findings relating to the study objectives. 1.5 pages should be good for well-structured discussions.Acknowledgement: though the authors mentioned that no financial assistance received for this study, however it will be appropriate to acknowledge the program that resulted in the data used in the study. Additionally, the 7 hospitals and the clients should be acknowledged.References: please review all the references and ensure they align with Plosone citation guidelines.Please submit your revised manuscript by Nov 15 2024 11:59PM. If you will need more time than this to complete your revisions, please reply to this message or contact the journal office at plosone@plos.org . Please include the following items when submitting your revised manuscript:

We look forward to receiving your revised manuscript.

Kind regards,

Ibrahim Jahun, MD, MSC, PhD

Academic Editor

PLOS ONE

Journal Requirements:

Reviewers' comments:

Reviewer's Responses to Questions

**Comments to the Author**

1. Is the manuscript technically sound, and do the data support the conclusions?

Reviewer #1: Yes

Reviewer #2: Yes

Reviewer #3: Partly

2. Has the statistical analysis been performed appropriately and rigorously? 

Reviewer #1: Yes

Reviewer #2: Yes

Reviewer #3: Yes

3. Have the authors made all data underlying the findings in their manuscript fully available?

Reviewer #1: Yes

Reviewer #2: Yes

Reviewer #3: No

4. Is the manuscript presented in an intelligible fashion and written in standard English?

Reviewer #1: Yes

Reviewer #2: No

Reviewer #3: Yes

5. Review Comments to the Author

Reviewer #1: 1. Among the participants who received enhanced adherence counselling, a significant

140 proportion of them had detectable VL compared with the ones who did not have detectable VL,

141 32.6% vs. 0.5%, p <0.001 (row 140 and 141, this sentence is standalone and not explanatory)

2, Row 142. A higher prevalence of the clients aged between 16 and 24 years had

detectable VL compared with those without detectable VL, 15.2% vs. 6.2% ( i suggest it can be a higher percentage or proportion rather than prevalence)

Reviewer #2: This paper addressed a very important topic. I have provided some detained comments in the attached documents to help improve the flow and readability. The authors can consider shortening the paper, restructuring the background and results section and discussing the key findings.

Reviewer #3: Short title has a minor grammatical error, remove 'of'. The inclusion and exclusion criteria is missing. There is need to show the age of consent and who needs assent. The Consent form was only in English, looking the population ,there was need to have a translated version to native language. Researcher may need to explain how the questionnaire was administered.

The data was not available in the manuscript

6. PLOS authors have the option to publish the peer review history of their article (what does this mean? ). If published, this will include your full peer review and any attached files.

**Do you want your identity to be public for this peer review?** For information about this choice, including consent withdrawal, please see our Privacy Policy .

Reviewer #1: No

Reviewer #2: No

Reviewer #3: **Yes: ** Sandra Shawarira-Bote

---

## [Author Response · Author response to Decision Letter 1]

28 Jan 2025

Comment

1.Data availability – stated “some restrictions will apply” please explain the level of restrictions and how the data can be accessed.

Response

Thank you.

In earlier submission the data had restrictions, however after consideration and appreciating the importance of the data in validation of the manuscript, the data has been uploaded on separate file (S1 minimal dataset)

Comment

2.Title: Prevalence and factors associated with detectable viral load among adult patients receiving ART in Choma District, Zambia. “Prevalence” is confusing in the title, is it referring to prevalence of factors or prevalence of detectable VL? Please make it clear, for example “Prevalence of Detectable VL and Associated Factors among Adult Patients Receiving ART in Choma District, Zambia”.

Response

Well noted thank you. The title was revised and shortened accordingly.

Comment

3.  Introduction: Good epidemiologic context provided.However, no reference made to factors influencing VL suppression. Consider restructuring the section using 95:95:95 cascade and about 50% of the section should focus on factors influencing VL suppression, including gaps in Zambia and study region.

Response

Thank you, the revision has been made to the introduction.

Comment

4.Methods:

oStudy design: is it primary data collection or secondary data analysis?

oSelection and sampling: 448 clients were selected. Please describe sampling criteria, method and sample size estimation. Also describe accurately “selection based on proportional to size”.

oData collection and storage: was the data collected using paper-based questionnaire or electronically? What are the variables of interest and their definition?

Response

Thank you, the section has been revised to and addresses all the concerns reviewers highlighted.

Comment

5.Results: not structured and presented without keeping the study objectives in mind. Of course, the study objectives were not well defined in the methods, and this may be the reason for the long and unstructured results section. The section is very lengthy and difficult to follow. No need to mention all findings. Focus on major findings that are elated to the objectives. Try to make the text for this section like maximum of 1 page. The tables should be self-explanatory. The tables are also too long and confusing. Focus on key findings and avoid repetition.

oTable 1: should be “Sociodemographic characteristics and VL suppression of participants from selected health facilities in Choma, Zambia”. The table should end at “perinatal HIV” because the variables that follow are not sociodemographic and should be moved to another table.

oTable 2: Please avoid repetition. Harmonize table 1 and 2 based on the comment above. Make Table 2 to focus on “VL suppression by site and clinical characteristics”

oTables 3 and 4: please harmonize these tables into 1 table and be concise by focusing on the primary intention of the table.  

Response

Well noted, Thank you. The revisions have been made accordingly

Comment

6.Discussion: too long. Please be concise and focus on key findings relating to the study objectives. 1.5 pages should be good for well-structured discussions.

Response

Thank you, the revision has been made accordingly.

Comment

7.Acknowledgement: though the authors mentioned that no financial assistance received for this study, however it will be appropriate to acknowledge the program that resulted in the data used in the study. Additionally, the 7 hospitals and the clients should be acknowledged.

Response

Thank you. The acknowledgments were revised.

Comments

8.References: please review all the references and ensure they align with Plosone citation guidelines.

Response

Thank you, this is well noted. The revision has been made according to provided guidelines.

Comment

9.Please ensure that your manuscript meets PLOS ONE's style requirements, including those for file naming

Response

Thank you. This has been done accordingly.

Comment

10.Reviewer #1: 

1. Among the participants who received enhanced adherence counselling, a significant

140 proportion of them had detectable VL compared with the ones who did not have detectable VL,

141 32.6% vs. 0.5%, p <0.001 (row 140 and 141, this sentence is standalone and not explanatory)

2. Row 142. A higher prevalence of the clients aged between 16 and 24 years had

detectable VL compared with those without detectable VL, 15.2% vs. 6.2% ( i suggest it can be a higher percentage or proportion rather than prevalence)

Response

Thank you, well noted. The revision has been made accordingly to make the information easier to understand.

Comment

11.Reviewer #2: This paper addressed a very important topic. I have provided some detained comments in the attached documents to help improve the flow and readability. The authors can consider shortening the paper, restructuring the background and results section and discussing the key findings.

Response

The comments in the attached document were well appreciated. Thank The revision has been made.

Comment

12.Reviewer #3: Short title has a minor grammatical error, remove 'of'. The inclusion and exclusion criteria is missing. There is need to show the age of consent and who needs assent. The Consent form was only in English, looking the population ,there was need to have a translated version to native language. Researcher may need to explain how the questionnaire was administered.

The data was not available in the manuscript

Comment

Thank you so much. The title has been shortened accordingly and the concerns addressed in the methods section.

---

## [Decision Letter · Decision Letter 1]

21 Feb 2025

Prevalence and factors associated with detectable viral load among adult patients receiving ART in Choma District, Zambia

PONE-D-24-18373R1

Dear Dr. Mutukwa,

We’re pleased to inform you that your manuscript has been judged scientifically suitable for publication and will be formally accepted for publication once it meets all outstanding technical requirements.

Kind regards,

Ibrahim Jahun, MD, MSC, PhD

Academic Editor

PLOS ONE

Additional Editor Comments (optional):

Reviewers' comments:

Reviewer's Responses to Questions

**Comments to the Author**

1. If the authors have adequately addressed your comments raised in a previous round of review and you feel that this manuscript is now acceptable for publication, you may indicate that here to bypass the “Comments to the Author” section, enter your conflict of interest statement in the “Confidential to Editor” section, and submit your "Accept" recommendation.

Reviewer #2: All comments have been addressed

Reviewer #3: All comments have been addressed

2. Is the manuscript technically sound, and do the data support the conclusions?

Reviewer #2: Yes

Reviewer #3: Yes

3. Has the statistical analysis been performed appropriately and rigorously? 

Reviewer #2: Yes

Reviewer #3: Yes

4. Have the authors made all data underlying the findings in their manuscript fully available?

Reviewer #2: Yes

Reviewer #3: Yes

5. Is the manuscript presented in an intelligible fashion and written in standard English?

Reviewer #2: Yes

Reviewer #3: Yes

6. Review Comments to the Author

Reviewer #2: (No Response)

Reviewer #3: (No Response)

7. PLOS authors have the option to publish the peer review history of their article (what does this mean? ). If published, this will include your full peer review and any attached files.

**Do you want your identity to be public for this peer review?** For information about this choice, including consent withdrawal, please see our Privacy Policy .

Reviewer #2: **Yes: ** Moses Bateganya

Reviewer #3: **Yes: ** Sandra Shawarira-Bote

---

## [Editor Report · Acceptance letter]

PONE-D-24-18373R1

PLOS ONE

Dear Dr. Mutukwa,

I'm pleased to inform you that your manuscript has been deemed suitable for publication in PLOS ONE. Congratulations! Your manuscript is now being handed over to our production team.

Kind regards,

on behalf of

Dr. Ibrahim Jahun

Academic Editor

PLOS ONE